# Drug resistance gene mutations and treatment outcomes in MDR-TB: A prospective study in Eastern China

**Qiao Liu**[1☯], **Dandan Yang**[2☯], **Beibei Qiu**[1☯], **Leonardo Martinez**[3], **Ye Ji**[1], **Huan Song**[1], **Zhongqi Li**[1], **Jianming Wang**[1]*

**1** Department of Epidemiology, Center for Global Health, School of Public Health, Nanjing Medical University, Nanjing, PR China, **2** Department of Sexually Transmitted Diseases and AIDS, Center for Disease Control and Prevention of Jiangsu Province, Nanjing, PR China, **3** Division of Infectious Diseases and Geographic Medicine, School of Medicine, Stanford University, Stanford, California, United States of America

☯ These authors contributed equally to this work.
* jmwang@njmu.edu.cn

## Abstract

### Background

Multidrug-resistant tuberculosis (MDR-TB) poses a serious challenge to TB control. It is of great value to search for drug resistance mutation sites and explore the roles that they play in the diagnosis and prognosis of MDR-TB.

### Methods

We consecutively enrolled MDR-TB patients from five cities in Jiangsu Province, China, between January 2013 and December 2014. Drug susceptibility tests of rifampin, isoniazid, ofloxacin, and kanamycin were routinely performed by proportion methods on Lowenstein–Jensen (LJ) medium. Drug resistance-related genes were sequenced, and the consistency of genetic mutations and phenotypic resistance was compared. The association between mutations and treatment outcomes was expressed as odds ratios (ORs) and 95% confidence intervals (CIs).

### Results

Among 87 MDR-TB patients, 71 with treatment outcomes were involved in the analysis. The proportion of successful treatment was 50.7% (36/71). The *rpoB* gene exhibited the highest mutation rate (93.0%) followed by *katG* (70.4%), *pncA* (33.8%), *gyrA* (29.6%), *eis* (15.5%), *rrs* (12.7%), *gyrB* (9.9%) and *rpsA* (4.2%). Multivariable analysis demonstrated that patients with *pncA* gene mutations (adjusted OR: 19.69; 95% CI: 2.43–159.33), advanced age (adjusted OR: 13.53; 95% CI: 1.46–124.95), and nonstandard treatment (adjusted OR: 7.72; 95% CI: 1.35–44.35) had a significantly higher risk of poor treatment outcomes.

**Data Availability Statement:** All relevant data are within the manuscript and its Supporting Information files.

**Funding:** This study was supported by the National Nature Science Foundation of China [81973103 (JW), 82003516(QL)], Medical Research Project of Jiangsu Health Commission (ZDB2020013)(JW), Postgraduate Research & Practice Innovation Program of Jiangsu Province (KYCX19-1131)(QL), and the Priority Academic Program Development of Jiangsu Higher Education Institutions (PAPD) (JW). The funders had no role in the study design, data collection and analysis, decision to publish, or preparation of the manuscript.

**Competing interests:** The authors have declared that no competing interests exist.

## Conclusions

These results suggest that *Mycobacterium tuberculosis* gene mutations may be related to phenotypic drug susceptibility. The *pncA* gene mutation along with treatment regimen and age are associated with the treatment outcomes of MDR-TB.

### Author summary

Multidrug-resistant tuberculosis (MDR-TB) exacerbates the already serious tuberculosis epidemic, poses a notable threat to global tuberculosis control, and places a considerable burden on developing countries, as treatments for MDR-TB tend to be expensive, of limited efficacy, and toxic. Genotypic determinants of resistance to specific drugs or drug classes offer a rapid and highly specific alternative to phenotypic drug susceptibility testing. Although the relationship between gene mutations and drug resistance has been described previously, the strength of the association of mutations with the treatment outcomes of MDR tuberculosis have not been fully elucidated. The results of our study, which was conducted in a Chinese population, suggest that gene mutations in *Mycobacterium tuberculosis* may be related to phenotypic drug susceptibility. Mutation of the *pncA* gene contributes to a poor prognosis and can be applied to predict the treatment outcomes of MDR-TB.

## Introduction

Tuberculosis (TB) remains a leading cause of death globally, particularly in low- and low-middle-income countries [1]. The increasing incidence of multidrug-resistant TB (MDR-TB), which is defined as resistance to at least isoniazid (INH) and rifampin (RIF), poses a serious challenge to TB control. In a nationwide survey across China, the prevalence of MDR-TB was 5.7% among new cases and 25.6% among previously treated cases [2].

The treatment of drug-resistant TB is usually more complex, toxic, and costly, as well as less effective, than the treatment of other forms of TB [3,4]. Both multidrug resistance and rifampin resistance alone are associated with an inadequate response to first-line treatment [5]. The critical regimens employed to treat MDR-TB include fluoroquinolones (FQs), second-line injectable anti-TB drugs (SLID), and pyrazinamide (PZA) [6,7]. Resistance to these drugs can lead to a prolonged treatment period and may increase the risk of unfavorable outcomes [7].

Gene mutations in *Mycobacterium tuberculosis (M.tb)* contribute to phenotypic drug susceptibility. The majority of mutations related to FQ resistance are located in the *gyrA* and *gyrB* genes [8]. Resistance to SLID is associated with gene mutations in rrs and eis, while resistance to PZA is associated with *pncA* and *rpsA* [9,10]. Recent studies have demonstrated that different mutations in *M.tb*, even those occurring within the same region, can confer different degrees of phenotypic resistance to anti-TB drugs [11–13]. Therefore, the combination of mutations at multiple sites has a comprehensive effect on drug resistance. We hypothesize that drug-resistant gene mutations can predict the drug-resistant phenotype and the prognosis of MDR-TB.

Although the relationship between gene mutations and drug resistance has been reported previously, the effect of common mutations on the treatment outcomes of MDR-TB has not been elucidated. In this study, we compared the frequency of mutations in common drug resistance-related genes with the *in vitro* drug susceptibility test (DST) results and evaluated their

roles in predicting the treatment outcomes of MDR-TB. In this study, we enrolled a group of MDR-TB patients and sequenced the mutation sites of the first- and second-line anti-TB drug-related genes in *M.tb*. The objective of this study was to investigate the relationship between gene mutations and the drug-resistant phenotype and determine its role in predicting the prognosis of MDR-TB.

## Methods

### Ethics approval and consent to participate

This study was approved by the Ethics Committee of Nanjing Medical University (approval number: 2019–225). Written informed consent was obtained from all participants. This study was conducted in accordance with the Declaration of Helsinki.

### Study subjects

A prospective cohort study was conducted in five cities in Jiangsu Province, China. Newly registered MDR-TB patients were enrolled from TB designated hospitals, including the Third People's Hospital of Changzhou, the People's Hospital of Taizhou, the Sixth People's Hospital of Nantong, the Infectious Disease Hospital of Xuzhou, and the Fourth People's Hospital of Lianyungang. MDR-TB was identified by regional reference laboratories using the traditional DST. Each patient signed an informed consent form followed by a questionnaire-based survey to gather demographic characteristics and clinical data. TB strains were isolated and transported to the Center for Disease Control and Prevention (CDC) of Jiangsu Province for verification and gene sequencing. Patients were followed for the treatment outcomes. Cure or completion of treatment was defined as treatment success. Death, treatment failure, discontinuation of treatment due to adverse reactions, or loss of follow-up were defined as unfavorable (poor) outcomes. Patients who achieved cured or completed treatment had a follow-up time ranging from 20 months to 26 months.

### Inclusion and exclusion criteria

Patients with MDR-TB detected by DST who could be followed up were eligible for the study. Those who were HIV-positive or had severe complications and were unable to participate in the study were excluded.

### Strain identification and DST

MTB was cultured and identified by p-nitrobenzoic acid (PNB) and thiophene carboxylic acid hydrazine resistance tests. Growth in Lowenstein-Jensen (LJ) medium containing PNB indicates that the bacilli do not belong to the MTB complex. Species other than MTB were excluded from the current analysis. DST was performed using a conventional proportion method on LJ medium according to the guidelines of the WHO. The LJ medium was impregnated with INH, RIF, kanamycin (KM), and ofloxacin (OFX). The concentrations of anti-TB drugs were 0.2 μg/ml for INH, 40 μg/ml for RIF, 30 μg/ml for KM and 2 μg/ml for OFX. The strain was defined as sensitive when the growth rate was < 1% compared to the control. Otherwise, the strain was declared resistant to the specific drug. For internal quality assurance of DST, a standard H37Rv strain was included in each new batch of the LJ medium.

### DNA sequencing of drug resistance-related genes

Mutations in the *rpoB*, *katG*, *inhA*, *pncA*, *rpsA*, *gyrA*, *gyr*B, *rrs*, and *eis* genes were analyzed by PCR amplification and then sequenced using the respective oligonucleotide primers

(S1 Table). Primers were designed by referring to previous studies[14–19]. Cycling conditions are shown in S2 Table. PCR products were sent to Sangon Biotech (Shanghai, China) for sequencing and aligned to the H37Rv reference strain (GenBank accession no. NC 000962) using ApE (v2.0.55, http://jorgensen.biology.utah.edu/wayned/ape/).

## Treatment regimens

The standard treatment of MDR-TB contains the intensive and continuation phases, ranging from 24 months to 27 months, depending on the length of the intensive phase [20]. Drugs used to treat MDR-TB included Pyrazinamide (PZA), Ethambutol (EMB), KM, Amikacin (Am), Capreomycin (Cm), OFX, Levofloxacin (Lfx), Moxifloxacin (Mfx), Cycloserine (Cs), Para-aminosalicylic acid (PAS), and Protionamide (Pto). Patients were treated with standardized or individualized regimens based on their treatment histories and DST results.

## Definitions

We defined new cases as patients who had never been treated for TB or had taken anti-TB drugs for less than one month. Previously treated patients were defined as those who had received one month or more of anti-TB drugs in the past. MDR-TB was defined as TB with resistance to at least both INH and RIF. Extensively drug-resistant TB (XDR-TB) was defined as TB with resistance to at least INH, RIF, KM, and OFX. Treatment outcomes were categorized according to the guidelines by the WHO [21]. Standard and individual treatment regimens were designed according to WHO guidelines [22,23]. Nonstandard treatment was defined as any MDR-TB treatment regimen that did not conform with the standard treatment regimen described previously. A "cured" patient was defined as one who had completed treatment according to the program protocol and had no evidence of treatment failure. Three or more consecutive cultures taken at least 30 days apart after the intensive phase should be negative. A "treatment completed" patient was defined as one who had completed treatment according to the program protocol but did not meet the definition of "cured" because of a lack of bacteriological results. The category of "died" comprised any patient who died for any reason during treatment. The "treatment failure" was recorded if the treatment was terminated or needed a permanent regimen change for at least two anti-TB drugs. The "lost to follow-up" category comprised any patient whose treatment was interrupted for two consecutive months or more. The category of "transferred out" comprised any patient transferred to another health facility and was unable to provide feedback on treatment outcomes. For the purpose of the analysis, we combined "cured" and "completed treatment" as "treatment success", whereas other outcomes were grouped as "poor treatment outcomes".

## Sample and data collection

Three sputum samples were collected from all suspected TB patients from regional TB designated hospitals in five cities followed by sputum smear detection and culture. Eligible MDR-TB patients were enrolled from TB designated hospitals and confirmed by a regional reference laboratory using traditional DST. MTB strains were isolated and transported to the Jiangsu CDC for cross-checking and gene sequencing. *M.tb* strains were stored for an extended time in a cryopreserved solution composed of tryptone and glycerol at -70˚C. Laboratory results, including sputum smear, culture, and DST results, were managed with the Jiangsu Provincial TB Laboratory Data Management System. Epidemiological data were input with EpiData 3.1 software (www.epidata.dk).

## Statistical analysis

We designed a questionnaire to collect demographic characteristics, disease history, treatment history, and behaviors of study subjects. We collected sputum smear, culture, and DST results from each TB reference laboratory. Treatment regimens and outcomes were recorded in the TB management system. After checking the questionnaire data, laboratory data, follow-up data, and sequencing data, a database was formed for analysis. Categorized variables were described using proportions. We used the logistic regression model to estimate the association and expressed it with odds ratios (ORs) and 95% confidence intervals (CIs). All statistical analyses were performed using SPSS 25.0 (IBM, NY, USA).

## Results

### Characteristics of study subjects

A total of 89 MDR-TB patients were recruited during the study period. After excluding the patients who could not meet the inclusion criteria, 71 patients were included in the final analysis (Fig 1). Among these patients, 24 (33.8%) were younger than 44 years, 52 (73.2%) were male, 29 (40.9%) were newly treated, 47 (66.2%) had an alcohol drinking history, 37 (52.1%) had a tobacco smoking history, and 48 (67.6%) were treated with the standard regimen recommended by the WHO. The proportion of successful treatment was 50.7%. Of the 35 patients with poor outcomes, 10 died, 8 had treatment failure, 13 had an adverse effect, and 4 were lost to follow-up (Table 1).

### Phenotypic drug resistance identified by DST

There were 20 (28.2%) strains resistant to OFX and 8 (11.3%) strains resistant to KM. A total of 51 (71.8%) isolates were only resistant to both INH and RIF, which was defined as simple MDR-TB. Seventeen (23.9%) MDR-TB isolates that were additionally resistant to either OFX or KM were defined as pre-XDR-TB, and 3 (4.2%) MDR-TB isolates that were additionally resistant to both OFX and KM were defined as XDR-TB.

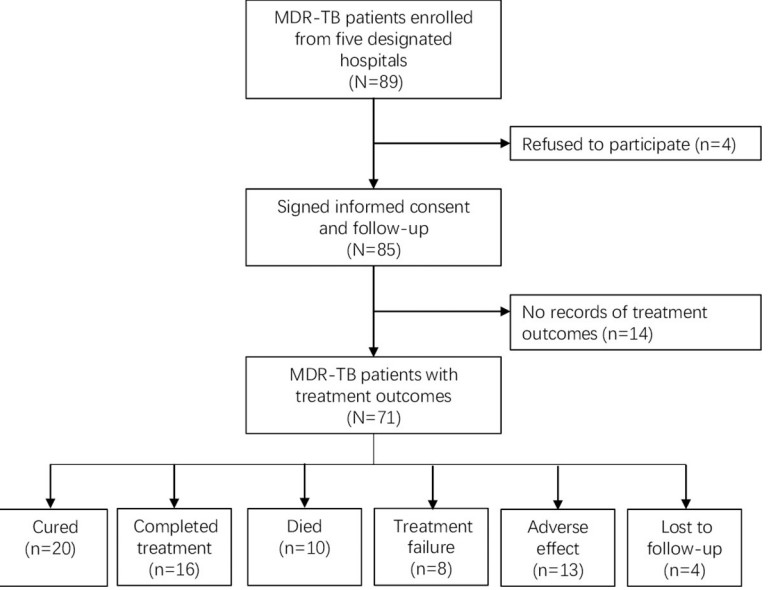

**Fig 1. Eligibility and enrollment of study subjects MDR-TB: multidrug-resistant tuberculosis.**

**Table 1. Demographic characteristics of 71 multidrug-resistant tuberculosis patients.**

| Variable | Total, N (%) | Treatment success, N (%) | Poor outcomes, N (%) |
|---|---|---|---|
| Sex | | | |
| Male | 52 (73.2) | 27 (75.0) | 25(71.4) |
| Female | 19 (26.8) | 9 (25.0) | 10 (28.6) |
| Age (years) | | | |
| <44 | 24 (33.8) | 15 (41.7) | 9 (25.7) |
| 44–54 | 25 (35.2) | 13 (36.1) | 12 (34.3) |
| >54 | 22 (31.0) | 8 (22.2) | 14 (40.0) |
| Weight (kg) | | | |
| <55 | 28 (39.4) | 10 (27.8) | 18 (51.4) |
| 55–60 | 23 (32.4) | 16 (44.4) | 7 (20.0) |
| >60 | 20 (28.2) | 10 (27.8) | 10 (28.6) |
| History of contact | | | |
| Yes | 6 (8.4) | 4 (11.1) | 2 (5.7) |
| No | 65 (91.6) | 32 (88.9) | 33 (94.3) |
| Smoking | | | |
| Yes | 37 (52.1) | 20 (55.6) | 17 (48.6) |
| No | 34 (47.9) | 16 (44.4) | 18 (51.4) |
| Drinking | | | |
| Yes | 47 (66.2) | 21 (58.3) | 26 (74.3) |
| No | 12 (16.9) | 7 (19.4) | 5 (14.3) |
| Missing | 12 (16.9) | 8 (22.2) | 4 (11.4) |
| Treatment history | | | |
| New cases | 29 (40.9) | 19 (52.8) | 10 (28.6) |
| Previously treated | 42 (59.2) | 17 (47.2) | 25(71.4) |
| Standard treatment | | | |
| Yes | 48 (67.6) | 30 (83.3) | 18 (51.4) |
| No | 23 (32.4) | 6 (16.7) | 17 (48.6) |
| Sputum smear grade | | | |
| <3+ | 40 (56.3) | 17 (47.2) | 23 (65.7) |
| ≥3+ | 31 (43.7) | 19 (52.8) | 12 (34.3) |

## DNA sequencing

The most prevalent mutations were located in *rpoB*531 (41/71; 62.1%) for RIF, *katG*315 (47/71; 66.2%) and C (–15) T in *inhA* (8/71; 11.3%) for INH; *gyrA*94 (9/71; 12.7%) for OFX; *rrs*1401 (4/71; 5.6%) and -22 in *eis* (10/71; 14.1%) for KM (Table 2). The proportion of mutations related to resistance to RIF, INH, PZA, OFX and KM was 93.0% (66/71), 77.5% (55/71), 38.0% (27/71), 38.0% (27/71), and 23.9% (17/71), respectively.

DNA sequencing identified 17 mutations in the *rpoB* gene from 66 (93.0%) isolates, with the most common one being at codon 531 (Ser→Leu) (41/71, 57.8%). Four types of mutations were identified in the *katG* gene from 50 (70.4%) isolates, with the most common one being at codon 315 (47/71, 66.2%). Double substitutions in *katG* and *inhA* were observed in 3 isolates. Among strains resistant to PZA, 24 (88.9%) carried mutations in *pncA*, and 3 (11.1%) carried mutations in *rpsA*. Among strains resistant to OFX, 21 (77.8%) carried mutations in *gyrA*, 7 (25.9%) carried mutations in *gyrB*, and 1 (3.7%) had mutations in *gyrA* and *gyrB*. Among strains resistant to KM, 9 (52.9%) had mutations in the *rrs* gene, and 14 (82.4%) had mutations in the *eis* gene (Table 2). Compared with DST, DNA sequencing exhibited a relatively good performance (S3 Table).

**Table 2. Genotypic and phenotypic characteristics of resistance to anti-tuberculosis drugs.**

| Drug | Gene | Locus | Nucleotide/codon change | Nucleotide/amino acid change | No of isolates |
|---|---|---|---|---|---|
| RIF | *rpoB* | 459 | CTG-CGG | LeT-Arg | 1 |
| | | 463 | CAG-CAC | Gln-His | 1 |
| | | 511 | CTG-CCG | LeT-Pro | 5 |
| | | 513 | CAA-CCA | Gln-Pro | 1 |
| | | 513 | CAA-CTA | Gln-LeT | 1 |
| | | 515 | ATG-CTG | Met-LeT | 1 |
| | | 516 | GAC-TAC | Asp-Tyr | 2 |
| | | 516 | GAC-GTC | Asp-Val | 5 |
| | | 522 | TCG-TTG | Ser-Let | 2 |
| | | 526 | CAC-TAC | His-Tyr | 2 |
| | | 526 | CAC-AAC | His-Asn | 3 |
| | | 526 | CAC-GAC | His-Asp | 2 |
| | | 526 | CAC-CGC | His-Arg | 2 |
| | | 526 | CAC-CTC | His-LeT | 3 |
| | | 526 | CAC-CAG | His-Gln | 1 |
| | | 531 | TCG-TTG | Ser-LeT | 41 |
| | | 533 | CTG-CCC | LeT-Pro | 2 |
| INH | *katG* | 241 | CCC-CGC | Pro-Arg | 1 |
| | | 289 | GAG-GGG | GlT-Gly | 1 |
| | | 291 | GCT-GTT | Ala-Val | 1 |
| | | 315 | AGC-ACC | Ser-Thr | 47 |
| | *inhA* | -15 | ACG-ATG | Thr-Met | 8 |
| PZA | *pncA* | 41 | GGT-GGA | Gly-Gly | 1 |
| | | 44 | GCA-GCC | Ala-Ala | 2 |
| | | 53 | CCG-CCA | Pro-Pro | 1 |
| | | 58 | TCT-CCT | Ser-Pro | 2 |
| | | 62 | AAT-AAG | Asn-Lys | 1 |
| | | 63 | GAT-GAC | Asp-Asp | 1 |
| | | 64 | AT_-ATC | _-Ile | 1 |
| | | 114 | CCA-CCG | Pro-Pro | 1 |
| | | 114 | CCA-CCC | Pro-Pro | 1 |
| | | 124 | TAC-GAC | Tyr-Asp | 1 |
| | | 127 | CGC-CGT | Arg-Arg | 1 |
| | | 155 | CGA-CGT | Arg-Arg | 1 |
| | | 164 | CCA-CCG | Pro-Pro | 1 |
| | | 187 | AGT-AGC | Ser-Ser | 2 |
| | | 192 | CGA-GGA | Arg-Gly | 1 |
| | | 195 | TCA-TCG | Ser-Ser | 1 |
| | | 198 | GCA-GCG | Ala-Ala | 1 |
| | | 201 | CAC-C_C | His- | 1 |
| | *rpsA* | 10 | _CC-CCC | _-Pro | 1 |
| | | 15 | AAT-AGT | Asn-Ser | 1 |
| | | 119 | AGA-ACA | Arg-Thr | 1 |
| | | 234 | _CC-CCC | _-Pro | 1 |
| | | 240 | _CC-CCC | _-Pro | 1 |
| OFX | *gyrA* | 45 | CTC-CTG | LeT-LeT | 1 |
| | | 60 | TTC-CTC | LeT-LeT | 2 |

(*Continued*)

**Table 2.** (Continued)

| Drug | Gene | Locus | Nucleotide/codon change | Nucleotide/amino acid change | No of isolates |
|------|------|-------|-------------------------|------------------------------|----------------|
| | | 90 | GCG-GTG | Ala-Val | 7 |
| | | 93 | ACG-ACC | Thr-Thr | 1 |
| | | 93 | ACG-ACT | Thr-Thr | 3 |
| | | 94 | ACA-GCA | Thr-Ala | 6 |
| | | 94 | ACA-CCA | Thr-Pro | 2 |
| | | 94 | ACA-ATA | Thr-Ile | 1 |
| | *gyrB* | 509 | AAC-ACC | Asn-Thr | 2 |
| | | 514 | GCG-GTG | Ala-Val | 1 |
| | | 522 | GGG-AGG | Gly-Arg | 4 |
| KM | *rrs* | 1205 | _-A | _-Met | 1 |
| | | 1227 | _-T | _-LeT | 1 |
| | | 1401 | A-G | Thr-Ala | 4 |
| | | 1449 | A-G | Arg-Gly | 1 |
| | | 1449 | T-C | Ser-Pro | 1 |
| | | 1601 | T-A | Ser-Arg | 1 |
| | *eis* | -57 | _-G | _-Ser | 1 |
| | | -54 | C-G | Asp-GlT | 2 |
| | | -22 | A-_ | Asn-_ | 10 |
| | | 2 | A-_ | Lys-_ | 1 |

RIF: rifampin; INH: isoniazid; PZA: promethazine; FQ: fluoroquinolone; KM; kanamycin.

### Factors related to treatment outcomes

As shown in Table 3, previous treatment history (OR: 2.79, 95% CI: 1.05–7.47, $P = 0.04$) and receiving a nonstandard treatment regimen (OR: 4.72, 95% CI: 1.55–14.17, $P = 0.01$) contributed to a poor treatment outcome. We also observed that *pncA* gene mutation (OR: 5.29, 95% CI: 1.76–15.89, $P < 0.01$), *pncA* gene mutation (OR: 5.29, 95% CI: 1.76–15.89, $P < 0.01$), and *gyrA* gene mutation (OR: 3.75, 95% CI: 1.25–11.30, $P = 0.02$) were related to a poor outcome (Table 4). We further explored the role of high-frequency mutation sites in the prognosis of MDR-TB but without significant findings (S4 Table).

After adjusting for potential confounders based on a multivariate logistic regression model, we observed that patients with advanced age ($\geq 54$ years) (adjusted OR: 13.53, 95% CI: 1.46–124.95, $P = 0.02$), receiving nonstandard treatment (adjusted OR: 7.72, 95% CI: 1.35–44.35, $P = 0.02$) and having mutations in the *pncA* gene (adjusted OR: 19.69, 95% CI: 2.43–159.33, $P < 0.01$) had an increased risk of poor treatment outcomes (Table 5).

### Discussion

MDR-TB poses a significant challenge to TB control. It takes several weeks for traditional DST to obtain results; therefore, it is of great value to search biomarkers for rapid detection and prediction of MDR-TB. A better understanding of the association between genotypic resistance to anti-TB drugs and treatment outcomes could facilitate the selection of effective drugs. We found that patients with advanced age, receiving nonstandard treatment, or infected with strains having mutations in PZA-related genes were at increased risk of poor treatment outcomes. PZA is a sterilizing drug with similar effectiveness as RIF. The *pncA* gene encodes pyrazinamidase, which converts PZA into its active form, pyrazinoic acid. Mutations in the *pncA*

**Table 3. Univariable logistic regression analysis of demographic factors associated with treatment outcomes.**

| Variable | Total, N (%) | Treatment success, N (%) | Poor outcomes, N (%) | cOR (95% CI) | *P* |
|---|---|---|---|---|---|
| Sex | | | | | |
| Male | 52 (73.2) | 27 (75.0) | 25(71.4) | Reference | |
| Female | 19 (26.8) | 9 (25.0) | 10 (28.3) | 1.20 (0.42–3.44) | 0.73 |
| Age (years) | | | | | |
| <44 | 24 (33.8) | 15 (41.7) | 9 (25.7) | Reference | |
| 44–54 | 25 (35.2) | 13 (36.1) | 12 (34.3) | 1.54 (0.49–4.81) | 0.46 |
| >54 | 22 (31.0) | 8 (22.2) | 14 (40.0) | 2.92 (0.88–9.67) | 0.08 |
| Weight (kg) | | | | | |
| <55 | 28 (39.4) | 10 (27.8) | 18 (51.4) | Reference | |
| 55–60 | 23 (32.4) | 16 (44.4) | 7 (20.0) | 0.24 (0.08–0.79) | 0.02 |
| >60 | 20 (28.2) | 10 (27.8) | 10 (28.6) | 0.56 (0.17–1.79) | 0.32 |
| History of contact | | | | | |
| No | 65 (91.6) | 32 (88.9) | 33 (94.3) | Reference | |
| Yes | 6 (8.4) | 4 (11.1) | 2 (5.7) | 0.49 (0.08–2.83) | 0.42 |
| Smoking | | | | | |
| No | 34 (47.9) | 16 (44.4) | 18 (51.4) | Reference | |
| Yes | 37 (52.1) | 20 (55.6) | 17 (48.6) | 0.76 (0.30–1.92) | 0.56 |
| Treatment history | | | | | |
| New cases | 29 (40.9) | 19 (52.8) | 10 (28.6) | Reference | |
| Previously treated | 42 (59.2) | 17 (47.2) | 25(71.4) | 2.79 (1.05–7.47) | 0.04 |
| Standard treatment | | | | | |
| Yes | 48 (67.6) | 30 (83.3) | 18 (51.4) | Reference | |
| No | 23 (32.4) | 6 (16.7) | 17 (48.6) | 4.72 (1.55–14.17) | 0.01 |
| Baseline sputum smear test | | | | | |
| <3+ | 40 (56.3) | 17 (47.2) | 23 (65.7) | Reference | |
| ≥3+ | 31 (43.7) | 19 (52.8) | 12 (34.3) | 0.47 (0.18–1.22) | 0.12 |

cOR: crude odds ratio, CI: confidence interval, TB: tuberculosis

gene are the dominant mechanism of drug resistance to PZA, which has been found in > 90% of PZA-resistant isolates [24,25].

In our study, half of the MDR-TB patients with PZA resistance were also at a high risk of resistance to FQs and SLID, indicating the role of the *pncA* mutation as a biomarker for MDR-TB. Similar findings have been observed in Chongqing, China, where PZA resistance was frequently accompanied by resistance to OFX, AM, KM, and CM [26], and *pncA* mutations were found in more than half of MDR-TB cases [27]. A study in South Africa showed that 69% of MDR-TB had *pncA* mutations, and 96% of XDR-TB had *pncA* mutations [28]. In a study in Shanghai, the mutation rate of the *pncA* gene was 37.8% in patients with MDR-TB [9]. PZA is used in susceptible TB treatment when used in combination with RIF, INH, and EMB and is a critical companion drug in new MDR-TB trials [29]. At present, there are few studies on the phenotype and genotype of PZA resistance in association with MDR-TB.

Correlation of drug resistance with a defect in a specific "drug-resistant" gene has been observed for INH, RIF, EMB, and OFX [30]. However, PZA appears to have multiple cellular targets, resulting in a variable correlation between PZA resistance and mutations in the *pncA* gene. Approximately 72–97% of PZA-resistant strains had *pncA* gene mutations. These mutated loci are numerous and scattered in the *pncA* gene [31]. In addition to *pncA*, other genes, such as *rpsA*, *panD*, and *hadC*, have also been reported to be related to PZA resistance,

**Table 4. Univariable logistic regression analysis of drug resistance gene mutations associated with treatment outcomes.**

| Variable | Total, N (%) | Treatment success, N (%) | Poor outcomes, N (%) | cOR (95% CI) | P |
|---|---|---|---|---|---|
| *rpoB* gene mutation | | | | | |
| No | 5 (7.0) | 2 (5.6) | 3 (8.6) | Reference | |
| Yes | 66 (93.0) | 34 (94.4) | 32 (91.4) | 0.63 (0.01–4.00) | 0.62 |
| *katG* gene mutation | | | | | |
| No | 21 (29.6) | 11 (30.6) | 10 (28.6) | Reference | |
| Yes | 50 (70.4) | 25 (69.4) | 25 (71.4) | 1.10 (0.40–3.05) | 0.86 |
| *inhA* gene mutation | | | | | |
| No | 63 (88.7) | 33 (91.7) | 30 (85.7) | Reference | |
| Yes | 8 (11.3) | 3 (8.3) | 5 (14.3) | 1.83 (0.40–8.34) | 0.43 |
| *katG* and *inhA* genes mutation | | | | | |
| No | 16 (22.5) | 9 (25.0) | 7 (20.0) | Reference | |
| Yes | 55 (77.5) | 27 (75.0) | 28 (80.0) | 1.33 (0.44–4.01) | 0.62 |
| *pncA* gene mutation | | | | | |
| No | 47 (66.2) | 30 (83.3) | 17 (48.6) | Reference | |
| Yes | 24 (33.8) | 6 (16.7) | 18 (51.4) | 5.29 (1.76–15.89) | <0.01 |
| *rpsA* gene mutation | | | | | |
| No | 68 (95.8) | 34 (94.4) | 34 (97.1) | Reference | |
| Yes | 3 (4.2) | 2 (5.6) | 1 (2.9) | 0.50 (0.04–5.78) | 0.58 |
| *pncA* or *rpsA* gene mutation | | | | | |
| No | 44 (70.4) | 28 (77.8) | 16 (45.7) | Reference | |
| Yes | 27 (29.6) | 8 (22.2) | 19 (54.3) | 4.16 (1.49–11.64) | <0.01 |
| *gyrA* gene mutation | | | | | |
| No | 50 (67.6) | 30 (83.3) | 20 (57.1) | Reference | |
| Yes | 21 (32.4) | 6 (16.7) | 15 (42.9) | 3.75 (1.25–11.30) | 0.02 |
| *gyrB* gene mutation | | | | | |
| No | 64 (90.1) | 32 (88.9) | 32 (91.4) | Reference | |
| Yes | 7 (9.9) | 4 (11.1) | 3 (8.6) | 0.75 (0.16–3.62) | 0.72 |
| *gyrA* and *gyrB* gene mutation | | | | | |
| No | 44 (62.0) | 26 (72.2) | 18 (51.4) | Reference | |
| Yes | 27 (38.0) | 10 (27.8) | 17 (48.6) | 2.46 (0.92–6.58) | 0.09 |
| *rrs* gene mutation | | | | | |
| No | 62 (87.3) | 30 (83.3) | 32 (91.4) | Reference | |
| Yes | 9 (12.7) | 6 (16.7) | 3 (8.6) | 0.47 (0.11–2.04) | 0.31 |
| *eis* gene mutation | | | | | |
| No | 60 (84.5) | 31 (86.1) | 29 (82.9) | Reference | |
| Yes | 11 (15.5) | 5 (13.9) | 6 (17.1) | 1.28 (0.35–4.66) | 0.71 |
| *rrs* and *eis* genes mutation | | | | | |
| No | 54 (76.1) | 26 (72.2) | 28 (80.0) | Reference | |
| Yes | 17 (23.9) | 10 (27.8) | 7 (20.0) | 0.65 (0.22–1.96) | 0.44 |

cOR: crude odds ratio; CI: confidence interval

but with inconsistency. Alexander believes that *rpsA* does not have a necessary ability to detect PZA resistance [32], while Wanliang notes that *rpsA* mediates relatively low-frequency mutations but can still detect PZA resistance [33]. A study in southern China found that a high-frequency mutation occurred at the 3' end of *rpsA* without *pncA* gene mutation, suggesting the value of *rpsA* in detecting PZA resistance [26]. The role of *rpsA*, *panD*, and *hadC* mutations with wild-type *pncA* genes in PZA resistance warrants further investigation [30].

**Table 5. Multivariable logistic regression analysis of drug resistance gene mutations and treatment outcomes.**

| Variable | Total, N (%) | Treatment success, N (%) | Poor outcomes, N (%) | aOR (95% CI) | P |
|---|---|---|---|---|---|
| Sex | | | | | |
| Male | 52 (73.2) | 27 (75.0) | 25(71.4) | Reference | |
| Female | 19 (26.8) | 9 (25.0) | 10 (28.3) | 2.14 (0.28–16.35) | 0.46 |
| Age (years) | | | | | |
| <44 | 24 (33.8) | 15 (41.7) | 9 (25.7) | Reference | |
| 44–54 | 25 (35.2) | 13 (36.1) | 12 (34.3) | 7.02 (0.87–56.62) | 0.67 |
| >54 | 22 (31.0) | 8 (22.2) | 14 (40.0) | 13.53 (1.46–124.95) | **0.02** |
| Weight (kg) | | | | | |
| <55 | 28 (39.4) | 10 (27.8) | 18 (51.4) | Reference | |
| 55–60 | 23 (32.4) | 16 (44.4) | 7 (20.0) | 0.18 (0.02–1.40) | 0.10 |
| >60 | 20 (28.2) | 10 (27.8) | 10 (28.6) | 0.40 (0.06–2.79) | 0.36 |
| History of contact | | | | | |
| No | 65 (91.6) | 32 (88.9) | 33 (94.3) | Reference | |
| Yes | 6 (8.4) | 4 (11.1) | 2 (5.7) | 5.24 (0.30–91.89) | 0.26 |
| Smoking | | | | | |
| No | 34 (47.9) | 16 (44.4) | 18 (51.4) | Reference | |
| Yes | 37 (52.1) | 20 (55.6) | 17 (48.6) | 0.66 (0.11–3.86) | 0.65 |
| Treatment history | | | | | |
| New cases | 29 (40.9) | 19 (52.8) | 10 (28.6) | Reference | |
| Previously treated | 42 (59.2) | 17 (47.2) | 25(71.4) | 0.75 (0.10–5.46) | 0.77 |
| Standard treatment | | | | | |
| Yes | 48 (67.6) | 30 (83.3) | 18 (51.4) | Reference | |
| No | 23 (32.4) | 6 (16.7) | 17 (48.6) | 7.72 (1.35–44.35) | **0.02** |
| Sputum smear grade | | | | | |
| <3+ | 40 (56.3) | 17 (47.2) | 23 (65.7) | Reference | |
| ≥3+ | 31 (43.7) | 19 (52.8) | 12 (34.3) | 0.34 (0.06–1.78) | 0.20 |
| *rpoB* gene mutation | | | | | |
| No | 5 (7.0) | 2 (5.6) | 3 (8.6) | Reference | |
| Yes | 66 (93.0) | 34 (94.4) | 32 (91.4) | 0.82 (0.01–53.17) | 0.93 |
| *katG* gene mutation | | | | | |
| o | 21 (29.6) | 11 (30.6) | 10 (28.6) | Reference | |
| Yes | 50 (70.4) | 25 (69.4) | 25 (71.4) | 0.39 (0.04–4.26) | 0.44 |
| *inhA* gene mutation | | | | | |
| No | 63 (88.7) | 33 (91.7) | 30 (85.7) | Reference | |
| Yes | 8 (11.3) | 3 (8.3) | 5 (14.3) | 0.67 (0.07–6.60) | 0.73 |
| *pncA* gene mutation | | | | | |
| No | 47 (66.2) | 30 (83.3) | 17 (48.6) | Reference | |
| Yes | 24 (33.8) | 6 (16.7) | 18 (51.4) | 19.69 (2.43–159.33) | **<0.01** |
| *rpsA* gene mutation | | | | | |
| No | 68 (95.8) | 34 (94.4) | 34 (97.1) | Reference | |
| Yes | 3 (4.2) | 2 (5.6) | 1 (2.9) | 0.96 (0.02–40.16) | 0.98 |
| *gyrA* gene mutation | | | | | |
| No | 50 (67.6) | 30 (83.3) | 20 (57.1) | Reference | |
| Yes | 21 (32.4) | 6 (16.7) | 15 (42.9) | 3.50 (0.512–23.94) | 0.20 |
| *gyrB* gene mutation | | | | | |
| No | 64 (90.1) | 32 (88.9) | 32 (91.4) | Reference | |
| Yes | 7 (9.9) | 4 (11.1) | 3 (8.6) | 0.49 (0.02–15.52) | 0.69 |

(*Continued*)

**Table 5.** (Continued)

| Variable | Total, N (%) | Treatment success, N (%) | Poor outcomes, N (%) | aOR (95% CI) | P |
|---|---|---|---|---|---|
| *rrs* gene mutation | | | | | |
| No | 62 (87.3) | 30 (83.3) | 32 (91.4) | Reference | |
| Yes | 9 (12.7) | 6 (16.7) | 3 (8.6) | 0.12 (0.01–1.78) | 0.12 |
| *eis* gene mutation | | | | | |
| No | 60 (84.5) | 31 (86.1) | 29 (82.9) | Reference | |
| Yes | 11 (15.5) | 5 (13.9) | 6 (17.1) | 0.93 (0.13–6.84) | 0.94 |

aOR: adjusted odds ratio; CI: confidence interval

FQs are considered the most critical component of MDR-TB treatment regimens, including OFX, Mfx, and Lfx. The DNA gyrase of *M.tb*, encoded by *gyrA* and *gyrB*, is well established as the quinolone target. The detection of mutations in the *gyrA* and *gyrB* genes has been demonstrated to be a promising technology for the rapid diagnosis of FQ resistance. In this study, we also observed that *gyrA* gene mutation was a risk factor for poor prognosis of MDR-TB, and the high-frequency mutation site was *gyrA*94. Treatment outcomes are primarily affected by the drug resistance level mediated by *gyrA* mutations. A study conducted by Harvard University found that high-level drug resistance caused by *gyrA* gene mutations was strongly associated with poor treatment outcomes. In contrast, the moderate-level resistance mediated by *gyrA* gene mutation did not show a significant statistical effect [34]. A multinational cohort study also suggested the role of high-level resistance-related *gyrA* mutations increased the risk of death for MDR/XDR-TB patients [35]. High doses of Mfx added to the MDR-TB treatment regimen can prevent patients from developing XDR-TB, indicating that low concentrations of FQ resistance caused by mutations in the *gyrA* locus can be treated with high doses of Mfx [36].

Although this study is the first to explore the genetic polymorphisms of dominant second-line anti-TB drug resistance genes in the treatment outcomes of MDR-TB in China, before generalizing the findings, we should not overlook its limitations. First, the sample size was relatively small, and these results must be confirmed in more extensive studies. Second, we did not perform MIC analyses to correlate gene mutations with the conferred resistance levels specifically. Third, we only conducted DST for INH, RIF, KM, and OFX and could not analyze the consistency between gene mutation and drug resistance of other anti-TB drugs. However, such biases were probably nondifferential with a tendency to diminish the strength of associations, rather than inflating them.

In conclusion, gene mutations in *M.tb* are related to phenotypic drug susceptibility. The *pncA* gene mutation, together with treatment regimen and age, can be applied to predict the prognosis of MDR-TB. Findings from this study suggest that drug-resistant gene testing should be conducted for MDR-TB patients to adjust the treatment regimens in a timely manner and improve the prognosis of patients. In China, new anti-TB drugs, such as bedaquiline and delamanid, are now being applied to treat MDR-TB. However, DST is not routinely performed to identify the drug resistance of these novel drugs. If molecular biology techniques can be widely utilized in clinical practice, they will help guide drug use and improve prognosis.

## Supporting information

**S1 Table. DNA sequencing primers for anti-tuberculosis drug resistance genes.**
(DOCX)

**S2 Table. Cycling conditions of anti-tuberculosis drug resistance genes.**
(DOCX)

**S3 Table. Performance of DNA sequencing compared to phenotypic DST.**
(DOCX)

**S4 Table. Univariate logistic analysis of high-frequency mutation sites and treatment outcomes.**
(DOCX)

## Author Contributions

**Conceptualization:** Qiao Liu, Dandan Yang, Beibei Qiu, Jianming Wang.

**Data curation:** Ye Ji, Huan Song, Zhongqi Li.

**Formal analysis:** Qiao Liu, Dandan Yang, Beibei Qiu.

**Funding acquisition:** Qiao Liu, Jianming Wang.

**Investigation:** Qiao Liu, Dandan Yang, Beibei Qiu, Ye Ji, Huan Song, Zhongqi Li.

**Project administration:** Leonardo Martinez, Jianming Wang.

**Supervision:** Leonardo Martinez, Jianming Wang.

**Validation:** Ye Ji, Huan Song, Zhongqi Li.

**Visualization:** Qiao Liu, Dandan Yang, Beibei Qiu.

**Writing – original draft:** Qiao Liu, Dandan Yang, Beibei Qiu.

**Writing – review & editing:** Qiao Liu, Leonardo Martinez, Jianming Wang.

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
