## [Decision Letter · Decision Letter 0]

16 Sep 2020

Dear Prof. Wang,

Thank you very much for submitting your manuscript "Drug Resistance Gene Mutations and Treatment Outcomes of MDR-TB: A Prospective Study in Eastern China" for consideration at PLOS Neglected Tropical Diseases. As with all papers reviewed by the journal, your manuscript was reviewed by members of the editorial board and by several independent reviewers. In light of the reviews (below this email), we would like to invite the resubmission of a significantly-revised version that takes into account the reviewers' comments. 

We cannot make any decision about publication until we have seen the revised manuscript and your response to the reviewers' comments. Your revised manuscript is also likely to be sent to reviewers for further evaluation.

Sincerely,

Abdallah M. Samy, PhD

Deputy Editor

Reviewer's Responses to Questions

**Key Review Criteria Required for Acceptance?**

**Methods**

-Are the objectives of the study clearly articulated with a clear testable hypothesis stated?

-Is the study design appropriate to address the stated objectives?

-Is the population clearly described and appropriate for the hypothesis being tested?

-Is the sample size sufficient to ensure adequate power to address the hypothesis being tested?

-Were correct statistical analysis used to support conclusions?

-Are there concerns about ethical or regulatory requirements being met?

Reviewer #1: Please describe sample collection fully, also how data was collected including the data collection software.

The objectives of the study are clear and the study design is appropriate. The study population is clearly described and appropriate for the hypothesis. The sample size is small and the authors acknowledged that.

There are concerns regarding data handling. No description on how samples were collected, stored and how data was collected and stored

Reviewer #2: The objectives of the study are clearly stated however, I did not see the author's hypothesis or maybe it is not well articulated. It is also not very clear how long the patients were followed; were they followed up until they completed their treatment?

The study population is described clearly and is appropriate for the hypothesis being tested. The authors need to clarify if patients with drug resistance other than MDR were excluded from the study i.e. what is the exclusion criteria? Appropriate statistical analyses were used to support the conclusions and all the ethical requirements were met.

**Results**

-Does the analysis presented match the analysis plan?

-Are the results clearly and completely presented?

-Are the figures (Tables, Images) of sufficient quality for clarity?

Reviewer #1: The analysis plan was not described in full. The results are clear and well presented and the figures are clear

Reviewer #2: The analysis presented match the analysis plan and the results are clearly presented. The tables are of sufficient quality for clarity and are very easy to understand.

**Conclusions**

-Are the conclusions supported by the data presented?

-Are the limitations of analysis clearly described?

-Do the authors discuss how these data can be helpful to advance our understanding of the topic under study?

-Is public health relevance addressed?

Reviewer #1: The conclusions are supported by the data represented and the limitations are outlined. It is not easy to judge the limitations because the methods were not described explicitly, including what sample was collected and how it was collected. The authors also do not fully address how the data will advance understanding of the topic and the public health relevance of the study

Reviewer #2: The conclusions are supported by the data that is presented by the authors and the limitations are duly noted. The authors need to state the relevance of their research to public health or rather clearly state how their data can be used to improve patient management/management of MDR TB cases. It would also be important for the authors to address their findings in the context of newer TB drugs for the treatment of MDR TB such as bedaquiline and delamanid. What do their findings mean for patients who are being treated with newer TB drugs?

**Editorial and Data Presentation Modifications?**

Reviewer #1: The citations in the paper are not done well and cannot be linked to the references. Authors should please familiarise themselves with their referencing style including doing citations

Reviewer #2: Authors need to clarify a few points that were raised before the paper is accepted for publication.

**Summary and General Comments**

Reviewer #1: The study is important in adding to the body of knowledge on drug resistant TB considering that the studies available have low numbers of participants.

Minor revision is needed

Reviewer #2: Overall, this manuscript is well written and addresses an important subject. The data presented in this manuscript improves our understanding of how drug resistance mutations affect treatment outcomes.

PLOS authors have the option to publish the peer review history of their article (what does this mean?). If published, this will include your full peer review and any attached files.

Reviewer #1: Yes: Goabaone Rankgoane-Pono

Reviewer #2: No
---

## [Decision Letter · Decision Letter 1]

28 Nov 2020

Dear Prof. Wang,

Thank you very much for submitting your manuscript "Drug Resistance Gene Mutations and Treatment Outcomes in MDR-TB: A Prospective Study in Eastern China" for consideration at PLOS Neglected Tropical Diseases. As with all papers reviewed by the journal, your manuscript was reviewed by members of the editorial board and by several independent reviewers. The reviewers appreciated the attention to an important topic. Based on the reviews, we are likely to accept this manuscript for publication, providing that you modify the manuscript according to the review recommendations. 

Sincerely,

Abdallah Samy

Deputy Editor

Reviewer's Responses to Questions

**Key Review Criteria Required for Acceptance?**

**Methods**

-Are the objectives of the study clearly articulated with a clear testable hypothesis stated?

-Is the study design appropriate to address the stated objectives?

-Is the population clearly described and appropriate for the hypothesis being tested?

-Is the sample size sufficient to ensure adequate power to address the hypothesis being tested?

-Were correct statistical analysis used to support conclusions?

-Are there concerns about ethical or regulatory requirements being met?

Reviewer #2: The objectives of the study are clearly stated and the study design is appropriate to address the objectives. The sample size is rather small however the authors have acknowledged this limitation. Authors have used the correct statistical tools which support their conclusions. This study was carried out without any ethical concerns, appropriate ethical clearance was obtained.

**Results**

-Does the analysis presented match the analysis plan?

-Are the results clearly and completely presented?

-Are the figures (Tables, Images) of sufficient quality for clarity?

Reviewer #2: The analysis presented matches the analysis plan and the results are clearly presented. However, the authors should clearly explain what they mean by nonstandard treatment (Line 260). The tables and figures are clear and easy to understand and interpret. Overall the results are well presented.

**Conclusions**

-Are the conclusions supported by the data presented?

-Are the limitations of analysis clearly described?

-Do the authors discuss how these data can be helpful to advance our understanding of the topic under study?

-Is public health relevance addressed?

Reviewer #2: Conclusion made are supported by the data that is presented and authors have clearly outlined the limitations of the study. The authors have discussed clearly how the data presented can help improve our understanding of TB drug resistance and the importance of checking for drug resistance mutations. the public health relevance of the study has been addressed; authors have stated how drug resistant gene testing can be used to improve patient treatment outcomes.

**Editorial and Data Presentation Modifications?**

Reviewer #2: Minor revision

**Summary and General Comments**

Reviewer #2: Overall the data that is presented in this manuscript is very important and relevant and helps us understand the impact of genetic mutations on treatment outcomes. This type of work is very important in this era of increased TB drug resistance and also offers an potential alternative to the usual phenotypic drug susceptibility testing.

PLOS authors have the option to publish the peer review history of their article (what does this mean?). If published, this will include your full peer review and any attached files.

Reviewer #2: No
---

## [Editor Report · Decision Letter 2]

12 Dec 2020

Dear Prof. Wang,

We are pleased to inform you that your manuscript 'Drug Resistance Gene Mutations and Treatment Outcomes in MDR-TB: A Prospective Study in Eastern China' has been provisionally accepted for publication in PLOS Neglected Tropical Diseases.

Best regards,

Abdallah M. Samy, PhD

Deputy Editor

---

## [Editor Report · Acceptance letter]

15 Jan 2021

Dear Prof. Wang,

We are delighted to inform you that your manuscript, "Drug Resistance Gene Mutations and Treatment Outcomes in MDR-TB: A Prospective Study in Eastern China," has been formally accepted for publication in PLOS Neglected Tropical Diseases.

Best regards,

Shaden Kamhawi

co-Editor-in-Chief

Paul Brindley

co-Editor-in-Chief
